# Current Concepts in the Diagnosis and Management of Adult Primary Immune Thrombocytopenia: Our Personal View

**DOI:** 10.3390/medicina59040815

**Published:** 2023-04-21

**Authors:** Tomás José González-López, Adrian Newland, Drew Provan

**Affiliations:** 1Hematology Department, Hospital Universitario de Burgos, 09006 Burgos, Spain; 2Academic Haematology Unit, Blizard Institute, Barts and the London School of Medicine and Dentistry, Queen Mary University of London, London E1 2BB, UK

**Keywords:** immune thrombocytopenia, autoimmune disease, diagnosis, clinical management, thrombopoietin receptor agonists, eltrombopag, avatrombopag, fostamatinib

## Abstract

Primary immune thrombocytopenia (ITP) is an acquired blood disorder that causes a reduction in circulating platelets with the potential for bleeding. The incidence of ITP is slightly higher in adults and affects more women than men until 60 years, when males are more affected. Despite advances in basic science, primary ITP remains a diagnosis of exclusion. The disease is heterogeneous in its clinical behavior and response to treatment. This reflects the complex underlying pathophysiology, which remains ill-understood. Platelet destruction plays a role in thrombocytopenia, but underproduction is also a major contributing factor. Active ITP is a proinflammatory autoimmune disease involving abnormalities within the T and B regulatory cell compartments, along with several other immunological abnormalities. Over the last several years, there has been a shift from using immunosuppressive therapies for ITP towards approved treatments, such as thrombopoietin receptor agonists. The recent COVID-19 pandemic has hastened this management shift, with thrombopoietin receptor agonists becoming the predominant second-line treatment. A greater understanding of the underlying mechanisms has led to the development of several targeted therapies, some of which have been approved, with others still undergoing clinical development. Here we outline our view of the disease, including our opinion about the major diagnostic and therapeutic challenges. We also discuss our management of adult ITP and our placement of the various available therapies.

## 1. Introduction

Adult primary immune thrombocytopenia (ITP) is an acquired autoimmune hematological disorder that leads to a reduced peripheral blood platelet count. This increases the patient’s risk of mucocutaneous bleeding and hemorrhage, with intracranial hemorrhage being the most serious complication observed in this disease [1,2].

Two types of ITP are recognized: primary and secondary ITP. Primary ITP, which accounts for around 80% of cases, is isolated thrombocytopenia not caused by or associated with another underlying disorder and is characterized by a platelet count < 100 × 10^9^/L. Secondary ITP includes all other immune-mediated thrombocytopenias, including autoimmune disorders (e.g., rheumatoid arthritis, Sjögren’s syndrome, and others) [3], lymphoproliferative disorders (e.g., CLL), drug-induced ITP (e.g., linezolid, or quinidine) and infections (e.g., HIV or HCV) [4]. In this review, we have restricted our discussion to adult primary ITP only.

The incidence of primary ITP in adults is 3.3–3.9 per 100,000 per year, with a slight female preponderance until the age of 60, when it is more common in males [5]. ITP is heterogenous in its clinical presentation and response to treatment [6].

The underlying cause of ITP and its pathophysiology is poorly understood. These are discussed later in the review. The platelet count reduction is due to a combination of increased platelet destruction by self-reacting B cells (autoantibodies) and T cells, in addition to a relative underproduction of platelets [7].

## 2. Confirming the Diagnosis of ITP

Despite clinical advances, the diagnosis of ITP remains one of exclusion. Patients with suspected primary ITP undergo several investigations to exclude underlying diseases such as haematinic deficiencies, marrow infiltration, other autoimmune diseases, viral infections, and other disorders. If no abnormalities are found, and the only finding is of isolated thrombocytopenia (platelets < 100 × 10^9^/L), a diagnosis of primary ITP is made. It is currently impossible to provide a positive diagnosis of ITP since there is no definitive diagnostic assay.

Historically, a bone marrow was often performed in patients presenting with thrombocytopenia to exclude bone marrow pathology. However, studies have shown a low pickup rate for serious bone marrow pathology in patients with no other abnormal clinical or laboratory signs. This was confirmed in a recent publication by Comont et al. [8]. For this reason, the latest International Consensus Guidelines and American Society of Hematology guidelines do not recommend routine bone marrow examination unless there are symptoms or signs suggesting the presence of an underlying disorder.

From real-world experience, we know that ITP may be incorrectly diagnosed, and the Hamilton group (Hamilton, ON, Canada) published useful data in 2017. Their recommendations for diagnosis are based on their routine clinical practice at the McMaster Centre [9]. In addition, they suggest a comprehensive and complete list of laboratory tests to avoid diagnostic errors. In our opinion, at present, we feel that the investigative recommendations of ICR and the ASH guidelines are sufficient to establish the diagnosis in the vast majority of cases [10,11].

### 2.1. New Tools for ITP Diagnosis

Several new tools are currently being developed for the diagnosis of ITP. There has been interest in using next-generation flow (NGF) in order to help make the diagnosis of primary ITP. We recently published our NGF data to differentiate ITP from myelodysplasia (MDS) using bone marrow and peripheral blood samples [12]. However, these analyses are not widely available or recommended for routine practice.

### 2.2. What Is the Role of Antiplatelet Antibody Testing?

ITP is mainly characterized by platelet and megakaryocytic destruction caused by anti-platelet autoantibodies present in the serum and bone marrow of the patient. These autoantibodies are observed free and also bound to platelets [3]. The sensitivity of anti-platelet antibodies in the serum is only around 40% but on the patient’s platelets is 50–80%. In the remaining cases, no antibodies are detected. However, the presence of platelet antibodies is not required to confirm the diagnosis of ITP. In addition, the presence or absence of antiplatelet antibodies or their subtypes does not correlate with treatment response. Thus, we do not support routine testing for anti-platelet antibodies, as treatment outcomes are similar regardless.

However, in patients where there is uncertainty about the immune cause of thrombocytopenia, a monoclonal antibody immobilization of platelet antigen (MAIPA) assay may be of help. Therefore, in our practice, we occasionally perform MAIPA assays, which has changed our clinical management. But these are not necessary for routine practice [13].

## 3. The Pathophysiology of ITP Is Complex and Poorly Understood

### 3.1. Cytokines and Th1

A number of studies have shown abnormalities in immune mediators, including cytokines and chemokines in ITP. It is broadly accepted that ITP is predominantly a proinflammatory TH1 disorder, with interferon gamma (IFNg) being the main mediator.

At diagnosis, there is usually an elevation of interleukin 17 (IL-17) caused by an increased TH17 response. This is probably due to a lack of T regulatory (Treg) suppression of autoimmunity [14]. However, there may also be defects in other cells, including natural killer cells and dendritic cells. Following successful therapy in patients where the platelet count returns to normal, IL-17 levels fall, and anti-inflammatory cytokines, including interleukin 10 and TGF-b, levels rise, indicating a shift from a TH1 (proinflammatory) state to an anti-inflammatory state.

### 3.2. T and B Reg Abnormalities

The role of T regulatory cells in ITP has been investigated fairly extensively, and in patients who respond to therapies such as rituximab, normalization in the number and function of Tregs seems to occur [15]. But Tregs are not the only abnormal cells in ITP, and the regulatory B cell (Breg) compartment also appears abnormal, allowing autoimmune disease to occur. The detailed underline pathophysiology is still poorly understood, and the interplay between Tregs, Bregs, and other cells along with cytokines is not fully understood. There are many detailed reviews on the pathophysiology of ITP, which will not be discussed here [7,16,17].

B regulatory (Breg) cells (CD19+CD24hiCD38hi) are involved in the pathophysiology of ITP by secreting IL10, a cytokine through which T regulatory (Treg) cells (CD4+CD25+Foxp3+) are recruited. Because there is a functional impairment of these Breg cells in chronic ITP, the amount of IL10 secreted by them is low, leading to decreased Treg cell recruitment and reduced Th cell function [18]. Furthermore, patients with active disease present with low levels of IL-10 compared to those in remission or healthy controls as Tregs, the cells that secrete IL-10 and IL-35, are clearly decreased in patients with ITP [18].

### 3.3. Cytotoxic T Cells

It has also been shown that cytotoxic T lymphocytes have a higher rate of proliferation and a lower rate of apoptosis in ITP, which leads to higher levels of IL-2 and IFNγ and lower levels of secreted IL10, alterations that are due to reduced levels and efficacy of Treg cells in patients with active ITP [18].

### 3.4. ITP and Complement

Activation of the classical pathway is implicated in the pathogenesis of ITP, with more severe disease occurring when low serum levels of C4 and/or C3 are accompanied by complement deposition on the surface of platelets [19,20]. Around 50% of ITP patients have antibodies that lead to complement activation or have detectable complement proteins on the platelet surface [20,21]. Thus, in some ITP patients, we can observe that specific IgG-antigen complexes can activate the C1 complex leading to the activation of the classical complement pathway generating C3a and C3b. The C3b deposited on platelets leads to subsequent platelet phagocytosis in the liver [19].

### 3.5. How Might Understand the Pathophysiology Help in the Development of Therapies?

In chronic ITP, there is an alteration in the Th1/Th2 ratio in favor of a Th1 (proinflammatory) response leading to an increase in Th1 precursors and the observation of a specific cytokine profile (very high IL-2 and IFNγ levels). To the best of our knowledge, no new clinical trial drug is intended to modify these cellular and cytokine alterations. However, recent data support the suggestion that some agents, such as thrombopoietin receptor agonists (TPO-RAs), may achieve a platelet response due, in part, to a modification of this cellular and humoral imbalance.

Spleen tyrosine kinase (Syk) is a non-receptor kinase expressed in a wide range of hematopoietic cells. Thus, Syk, as a key molecule in innate and adaptive immunity, plays an essential role in platelet phagocytosis. In addition, downstream signaling of various types of receptors such as FcγR, CR3, Dectin-1, and the apoptotic cell-recognizing receptor is mediated by Syk. Other functions of Syk in immunoreceptor- or integrin-mediated signaling in hematopoietic cells are still under investigation [22]. However, it has taken more than 15 years to clinically use different molecules, e.g., fostamatinib, which can prevent the destruction or phagocytosis at the splenic level of platelets opsonized with autoantibodies [23].

### 3.6. Increasing Bone Marrow Platelet Production and the Development of the Second Generation Thrombopoietin Receptor Agonists (TPO-RAs)

Harrington described platelet destruction many years ago as a cause of thrombocytopenia in ITP [24], but clearly, platelet underproduction is a major contributory factor [17,25,26].

Megakaryopoiesis and platelet production take place in the bone marrow (BM). Megakaryopoiesis involves molecular and cellular changes initiated and regulated by thrombopoietin (TPO), produced in the liver. TPO is responsible for the differentiation of hematopoietic stem cells (HSCs) into megakaryocytes that ultimately release platelets into the bloodstream [1].

Since stimulation of megakaryopoiesis with thrombopoietin mimetics leads to a platelet increase, three second-generation TPO-RAs (romiplostim, eltrombopag, avatrombopag) have been licensed for primary ITP. These TPO-RAs act by stimulating platelet production and thereby inducing responses in ≥80% of treated patients. Recent guidelines consider TPO-RAs as the main drug class for second-line therapy when patients fail to respond, relapse after the initial response, or become corticosteroid-dependent [10,11].

## 4. Clinical Management of ITP

Given the growing understanding of the pathophysiology of ITP, and as novel mechanisms involved in the development of the disease have been discovered, several drugs with alternative molecular targets that may show efficacy in ITP have been explored. As previously mentioned, we consider the TPO-RAs to be the most effective and safe drugs approved for second-line ITP [10,27]. In addition, our current therapeutic approach to ITP has been published recently [27].

There are numerous clinical guidelines that provide guidance on the treatment of ITP. Two major recently published guidelines are the Updated International Consensus Guidelines [10] and the ASH ITP guidelines [11]. Both guidelines agree on many of the recommendations but diverge on others. Based on the data and guidance published in these guidelines, we present our personal view on the treatment of ITP in the first line of treatment and beyond.

In addition, several drugs that will soon provide us with new treatment tools are currently in the clinical trial phase with clearly encouraging results. Therefore, it is likely that ITP treatment in the coming years will be quite different from today and that our view will change in a short period [28]. Who knows?

In line with the International Consensus Guidelines [10] and the ASH guidelines [11], we agree that the first-line treatment of ITP should remain corticosteroids (dexamethasone, prednisone, methylprednisolone) with or without intravenous immunoglobulin (IVIg) but limiting the use of first-line corticosteroids for 6 weeks or less in order to decrease unwanted effects and also the negative impact on quality of life.

Corticosteroids achieve long-term treatment-free remissions in only about 20% of patients, which means that 80% of patients will require second-line therapy [29]. Despite the excellent results obtained with dexamethasone in first-line therapy, in our hospital, we continue to use prednisone as the corticosteroid of choice for first-line treatment of ITP [27]. We use 1 mg/kg/day of prednisone (or prednisolone) for 1 week. If no response is seen after that first week, a second week of treatment may be administered at 1.5 mg/kg/day. If there is no platelet response after this second week, we advise the corticosteroids to be tapered and stopped [27].

IVIg 10% is as effective as IVIg 5% as an initial treatment for newly diagnosed adult ITP patients. Thus, IVIg achieves response rates ranging from 72.2% to 80.7% [10]. We reserve IVIg in ITP for bleeding patients or those at high risk of bleeding. In our Department in Spain, the dose administered (2 g/kg body weight) follows this schedule for 2 days [27]. The UK guidance is for 1 g/kg for one day only. We only use platelet transfusions in ITP where there is life-threatening bleeding or a high risk of bleeding. In emergency situations, we may use romiplostim starting at 10 µg/kg/week as a therapeutic alternative to IVIg [27]. We agree with current guidelines recommending early use of steroid-sparing agents such as TPO-RAs or fostamatinib that have demonstrated better disease outcomes and improved quality of life [10,11,23,27,30,31,32].

For second-line treatment, we base our decisions on clinical practice guidelines, but most importantly, we take into account the patient’s preferences. The degree of anxiety, the type of treatment (chronic or not), and/or its mode of administration (intravenous/subcutaneous or oral) also influence the treatment we choose. The significant percentage of ITP patients who wish to avoid chronic treatment and women’s reproductive desire also determine our treatment strategy, as many of the drugs commonly used in ITP are not approved for use during pregnancy.

For adults with ITP who are corticosteroid-dependent or unresponsive, there are numerous treatment options available. TPO-RAs are most commonly used as second-line therapy in ITP, and their mechanism of action is to stimulate megakaryopoiesis, thereby increasing platelet production [30,31,32]. Fostamatinib, on the other hand, is well suited for patients with high cardiovascular risk or a history of thrombosis [23]. The selection of the most appropriate TPO-RA or fostamatinib to treat ITP depends mainly on patient preference, as each drug has its own advantages and disadvantages.

Romiplostim and eltrombopag have been approved by the US Food and Drug Administration (FDA) and the European Medicines Agency (EMA) for the treatment of adults with ITP who have had an insufficient response to corticosteroids, IVIg or splenectomy. In addition, Avatrombopag was approved by the FDA in June 2019 for adult patients with chronic ITP who have had an insufficient response to previous treatment [33]. Other TPO-RAs are currently under investigation.

Romiplostim is administered at an initial dose of 1 µg/kg per week subcutaneously. However, in routine clinical practice, the usual starting dose is higher (3 µg/kg), and dose adjustments up to 10 µg/kg per week are made depending on platelet count response [30]. Response rates of 74% to 96% have been reported with its use [10,30].

Eltrombopag is administered at a starting dose of 25 or 50 mg/day, depending on the age and race (Asian or non-Asian) of the patient and the presence or absence of liver disease, up to a maximum dose of 75 mg/day [31,32]. Similarly to romiplostim, eltrombopag attained elevated response rates ranging from 50% to 88.8% [10,31,32]. Avatrombopag is initially administered as a 20 mg tablet with dose increments up to 40 mg/day depending on platelet counts, with platelet response rates of 93% in adult ITP. Avatrombopag, unlike eltrombopag, does not interact with calcium-rich foods [33].

TPO-RAs may be associated with the development of bone marrow fibrosis [30,31,32]. Eltrombopag retains the “caveat” of being associated with hepatotoxicity and can potentially decompensate patients with previous liver disease [31,32].

After the abrupt discontinuation of TPO-RAs, rebound thrombocytopenia has been reported [31,32]. However, some recent publications suggest that a percentage of patients (not yet clearly established but probably up to 30%) [34] are able to successfully discontinue this type of drug, leading to *sustained remission off-treatment* (SROT) while maintaining a hemostatic baseline platelet count without requiring additional treatment for ITP [34].

Our recommendation is to use any of the three TPO-RAs (eltrombopag, romiplostim or avatrombopag) as second-line treatment [27]. If, after the use of one TPO-RA, it is necessary to discontinue it due to refractoriness (non-response) or grade 3–4 serious adverse events (SAEs), any of the other two TPO-RAs can be chosen [27,31,32] since the efficacy rates of the three TPO-RAs are similar and adverse effects are most likely to remit after switching [35]. Our overall recommendation is to try all three TPO-RAs before moving on to another therapy since the TPO-RAs are the treatments with the highest efficacy to date.

Al-Samkari et al. have recently published data on TPO-RA switching, and in particular, switching to avatrombopag [36]. The authors emphasize that “switching” any TPO-RA to avatrombopag is probably the best possible therapeutic alternative for switching in ITP, especially when a response has not been achieved with the first TPO-RA used [36]. Thus, when it is necessary to consider switching between TPO-RAs due to the lack of efficacy of a first TPO-RA, we will preferably choose avatrombopag as the second TPO-RA [27]. We would, however, avoid a TPO-RA if the reason for stopping was bone marrow fibrosis; in this setting, we would choose a different class of drug.

To date, we do not have extensive experience with the use of avatrombopag, but our opinion of its use is very positive. We consider the non-interaction of avatrombopag with food to be its main advantage. On the other hand, our limited experience with its use restricts our comparison with the other TPO-RAs.

Fostamatinib belongs to another group of drugs with proven efficacy in the treatment of ITP: the splenic tyrosine kinase (Syk) inhibitors. This drug prevents phagocytosis of antibody-opsonized platelets by the spleen [23]. Fostamatinib, at an initial dose of 100 mg twice daily (frequently increased to 150 mg twice daily in nonresponders), achieved an overall response (platelets ≥ 50 × 10^9^/L) rate of 43% but with a median stable response (platelets ≥ 50 × 10^9^/L for 4 of 6 weeks) of only 18% [23]. Similarly, a 17% stable response rate was observed in patients previously treated with a TPO-RA [23]. Most adverse events (AEs) with fostamatinib are mild to moderate and are easily managed with dose reduction or discontinuation. Hypertension and diarrhea are the most frequent of them [23]. In addition, Fostamatinib is associated with a very low rate of thromboembolic events and has no immunosuppressive effect [23]. For these reasons, we consider the use of fostamatinib as a useful second-line therapy.

Our current clinical practice using TPO-RAs has recently been published in a protocol called “The Burgos Protocol” [27]. In the Burgos Protocol, we recommend tailoring our therapy to the clinical characteristics of the patient, with TPO-RAs and fostamatinib as major drugs to be used as second and subsequent lines of therapy.

For patients with chronic ITP (ITP longer than 12 months since disease diagnosis), splenectomy remains a valid therapeutic option. The early response rate to splenectomy is approximately 80%, arguably the only curative procedure for this disease. However, splenectomy is associated with complications such as thrombosis and/or infections due to encapsulated bacteria [10,11]. Therefore, in our clinical practice, we reserve splenectomy for patients who do not achieve a response after at least three second-line drugs: two TPO-RAs (any three of them) and fostamatinib [27]. This, coupled with the fact that most of our patients tend to avoid major surgery, keeps our splenectomy rate very low at around 4%.

According to the guidelines of the Spanish ITP Group (GEPTI) [37], the use of immunosuppressants (including the use of rituximab) in times of pandemics, e.g., COVID-19 pandemic, should be avoided due to the potential for serious infections caused by SARSCoV-2. Nevertheless, this recommendation is controversial because recent data showed no evidence of adverse effects when rituximab was used during the pandemic, with no change in hospitalization, intensive care unit admission or death rates following its use [38].

Outside this pandemic, current Spanish guidelines recommend these drugs as third-line treatment options after corticosteroids, intravenous immunoglobulins, thrombopoietin analogs (TPO-RAs; romiplostim, eltrombopag, avatrombopag) and Syk inhibitors (fostamatinib) [39].

The other immunosuppressant agents, including rituximab, MMF, ciclosporin and others, have proved useful for many patients over the years. However, their lack of approval and randomized data for ITP, side effects, and variable efficacy in monotherapy (63%, 57%, 55% and 52% of remissions with danazol [40], rituximab [41], dapsone [42] and mycophenolate mofetil [43], respectively), and immunosuppressive nature make them less attractive, especially with the introduction of more effective and well-tolerated treatments. Nonetheless, these drugs should be explored and discussed with patients [27].

### New Classes of Treatment Are in the Pipeline

Although the TPO-RAs have been successful for many patients, there remains a need for new classes of treatment. There are currently new drugs in clinical trials (efgartigimod, rilzabrutinib, sutimlimab and others) that should be approved for use in ITP in the next few years. As mentioned above, it is likely that data from these clinical trials will soon change our view of ITP treatment [28].

## 5. Unmet Needs in ITP

In our opinion, there are three major unmet needs in ITP that need to be addressed to better understand its pathophysiology.

The diagnosis of ITP remains a major challenge. Although many diagnostic tools are currently available, we still do not have a diagnostic test for ITP.Standardization of ITP into different pathological groups or risk groups is still impossible. We need to develop tools to identify those who need treatment and those who may avoid it. It would be clinically useful to know the predominant abnormality in an individual patient so that the patient may be offered treatment likely to be effective.We do not yet know which patients will respond to each of the different treatments for ITP currently available to us. Nor do we know which pathogenic mechanism is involved in the ITP of each of our patients: is it primarily humoral immunity, cell-mediated immunity, complement, platelet desialylation, or a combination of all of these? The underlying pathophysiology most likely varies from patient to patient, but we lack the assays to detect this.

## 6. Conclusions/“Take Home Messages”

New high-efficacy drugs, including avatrombopag and fostamatinib, have recently been developed that increase our treatment options for this disease.

However, despite the publication of several national and international guidelines, ITP remains a disease in which a high level of expertise is necessary for proper management. Therefore, in this review, we describe our personal view on the diagnosis and treatment of ITP.

The main points we would like to highlight are as follows:-For ITP diagnosis, we recommend following the diagnostic recommendations of the IWG [10]. In addition, physicians might consider referring ITP patients (at least for a good initial diagnostic evaluation) to better-equipped centers.-In our experience, we have been able to differentiate ITP from MDS reasonably well at diagnosis using next-generation flow cytometry (NGF) techniques on peripheral blood and bone marrow samples. However, we consider that it is still too early to establish the use of NGF as a “standard” diagnostic tool in our ITP patients.-Refractory ITP remains the biggest challenge in the management of the disease. We still need updated definitions, but this is being addressed by a consensus group that is updating the older 2009 consensus [44].-The role of TPO-RAs and their immunomodulatory effect that may lead to early immunomodulation of the disease makes the use of this class of treatment highly attractive in combination with steroids [45,46]. TPO-RAs have an obvious potential to affect the course of the disease [47,48,49], but until further studies are done, we suggest keeping the first-line treatment of ITP as indicated in IWG guidelines [10] and in our protocol [27]—corticosteroids ± IVIG.

We will probably soon have an answer to a question for which we currently lack precise data: could early use of TPO-RAs be the cure for ITP?

-Patients with second-line ITP should be treated on a case-by-case basis, and treatment should be individualized according to the patient’s comorbidities and thrombotic risk. Although it is unclear whether TPO-RAs may increase thrombotic risk in our patients [50,51,52,53,54], we recommend the use of fostamatinib in patients at high thrombotic risk [55,56,57]. Still, we recognize that rituximab and MMF also play an important role in the management of many patients.-Our recently published Burgos protocol proposal describes how to initiate treatment with TPO-RAs and how to switch these drugs [27]. Given their great results, we only recommend avatrombopag to rescue patients who have demonstrated a lack of efficacy of eltrombopag and/or romiplostim [36]. However, when switching is necessary due to adverse events, any of the three TPO-RAs currently approved in Europe are excellent alternatives [58].-The use of rituximab has declined in recent years [27], especially during this COVID-19 pandemic, in which the use of immunosuppressants is undesirable [37].-We only recommend splenectomy in patients who do not respond to at least three drugs: two of the three TPO-RAs available in Europe and fostamatinib, again with full discussion and patient agreement [59]. Or, if discontinuation of TPO-RAs is unsuccessful and the patient does not wish to be on continuous TPO-RA treatment much longer, we may also recommend splenectomy.

## Data Availability

No new data were created or analyzed in this study. Data sharing is not applicable to this article.

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
