# Peer review of "Current Concepts in the Diagnosis and Management of Adult Primary Immune Thrombocytopenia: Our Personal View"

_medicina, 2023, doi:10.3390/medicina59040815_

Round 1
Reviewer 1 Report
The manuscript represents a personal view of three experts on the management of ITP. The review describes the pathophysiology and diagnosis and covers modern treatment of ITP.
Following are my comments:
1- The statement regarding the NGF is misleading. It gives the impression that we need to do NGF in all patients which is not the case. Any flow would differentiate between MDS and ITP. Please clarify this point.
2- Another statement needs to be revised is the one concerning antiplatelet antibodies. We do not recommend measuring anti-platelet antibodies not because the results do not impact the treatment results, but rather because they do not have a clear diagnostic value. Please rephrase.
3- In addressing the pathophysiology, the authors use the term “chronic ITP” several times. Do you mean ITP> 12 m or adult ITP in general?
4- Page 3, line 113: the rate of durable response with TPO is around 65% according ASH guidelines. Please correct
5- Page 3, line 114-120. The authors refer to the Burgos protocol. This is something unknown to me and I am sure to the majority. Please omit or rephrase this statement
6- If fostamatinib prevents the destruction of platelets or phagocytosis at the splenic level, then how would you explain its efficacy in splenectomized patients?
7- Page 4, line 126: TPO-RA and fostamatinib are the only two 2nd line medications approved for ITP, however, fostamatinib is definitely not among the most effective and/or safest 2nd line medications. Please rephrase.
8- Page 4, line 134: unnecessary to refer to Dr. Drew Provan here. Please delete “coordinated by Dr. Drew Provan»
9- Page 4, line 154: Corticosteroids provide remissions in 20-30%. I believe 20% is a too low.
10- Page 4, line 164: we recommend limiting the use of cs to 6-8 weeks. That is not primarily because of a negative impact on QoL because we do not know that! I would rather say to limit side effects (short and long term).
11- Page 5: Is there any data showing that romiplostim 10ug is as effective and as quick as IVIG?
12- Page 5 line 159. There are no data to support the notion that early use of fosta improves Qol, neither do for TPO. Please revise this statement.
13- Page 5, line 172: The problem with choosing fosta in patients who fail to respond adequately to corticosteroids and in particular those with high risk of thromboembolism is not always possible because of the limitations imposed by the label (chronic ITP). This has to be clarified. Also why not rituximab to those with high risk of thromboembolism. Please discuss
14- The authors acknowledge that TPO may cause myelofibrosis. What is their approach/practice to follow up the patients on TPO to discover myelofibrosis.
15- Page 6, I cannot see how the 7-8 line describing the Spanish eltrombopag registry and the publications generated from that registry is relevant to the context of the manuscript. The whole passage can be deleted.
16- Page 6, line 209: The statement concerning switching to another TPO because of toxicity is not always true. You will not switch to another TPO if the reason was MF. You switch if the AE is related to the drug but you do not do so if the AE is a class effect. This has to be clarified.
17- Avatrombopag was recently licensed in Spain…) has unnecessarily been repeated several times.
18- Page 6, line 230. As mentioned earlier, I would say, there are indications that fosta is effective when used as second line since the evidence was based on post-hoc analysis.
19- Page 6, line 240: please add the rate “rate of XX%” !
20- I disagree with the author concerning their opinion on rituximab and in particularly when compared with fosta. Rituximab is more effective than fosta and much better tolerated. I am sure if we had conducted a head-to-head comparison, rituximab would have surpassed fosta in effect, safety and QoL and of course cost-effectiveness. This may also be applies to MMF. The British has extensive experience with MMF. Please moderate your statements regardless of what the Spanish guideline es says.
21- Unmet needs, first point. I feel there is a mix between diagnostic and prognostic markers. Pls re-phrase to clarify.
22- The statement regarding the use of rituximab during the covid was contraindicated is “outdated” and we know now is incorrect. We all know now that rituximab did not increase the risk of acquiring COVID infection neither did it complicate the course of disease. Even it is impact on suppression of humoral response to vaccine is of unknown clinical value!
Author Response
|
ITP, Our Personal View |
|
REVIEWER #1 |
|
|
|
Comments and Suggestions for Authors |
|
The manuscript represents a personal view of three experts on the management of ITP. The review describes the pathophysiology and diagnosis and covers modern treatment of ITP. |
|
Following are my comments: |
|
|
|
1- The statement regarding the NGF is misleading. It gives the impression that we need to do NGF in all patients which is not the case. Any flow would differentiate between MDS and ITP. Please clarify this point. |
|
Thank you for this comment. We agree the NGF section is misleading and we have reworded this. |
|
|
|
2- Another statement needs to be revised is the one concerning antiplatelet antibodies. We do not recommend measuring anti-platelet antibodies not because the results do not impact the treatment results, but rather because they do not have a clear diagnostic value. Please rephrase. |
|
Thank you for this comment. We agree with the reviewer and we have rephrased this section. |
|
|
|
3- In addressing the pathophysiology, the authors use the term “chronic ITP” several times. Do you mean ITP> 12 m or adult ITP in general? |
|
We agree this is confusing. Wherever the term “chronic” is used we mean ITP of 12 months or longer but the article predominantly deals with adult primary immune thrombocytopenia. |
|
|
|
4- Page 3, line 113: the rate of durable response with TPO is around 65% according ASH guidelines. Please correct |
|
Thank you for this comment. The response rates and durability of response varies from study to study and from drug to drug. For example, the durable response for avatrombopag is around 35% Romiplostim may be slightly higher. We have tried to clarify this within the manuscript. |
|
|
|
5- Page 3, line 114-120. The authors refer to the Burgos protocol. This is something unknown to me and I am sure to the majority. Please omit or rephrase this statement |
|
Thank you for this comment. The Burgos protocol is one which was published recently and is a Spanish protocol. We have added some text to expand on exactly what this is because, as the reviewer states, many clinicians will not have heard of this. |
|
|
|
6- If fostamatinib prevents the destruction of platelets or phagocytosis at the splenic level, then how would you explain its efficacy in splenectomized patients? |
|
This is an interesting point. In terms of efficacy in splenectomised patients this probably reflects the fact that platelet destruction involves the reticuloendothelial system as a whole which would include lymph nodes, bone marrow, lungs and other sites, as well as the spleen. |
|
|
|
7- Page 4, line 126: TPO-RA and fostamatinib are the only two 2nd line medications approved for ITP, however, fostamatinib is definitely not among the most effective and/or safest 2nd line medications. Please rephrase. |
|
We agree with the reviewer here. The most effective and safe second line medications are TPO-RAs. Following failure of three TPO-RAs we would probably look to rituximab or MMF, or fostamatinib in some cases. Fostamatinib efficacy is around 40% and it does have some side effects so amongst the authors there is some variation in where we place the drug but I would say fostamatinib is probably more of 3 line drug onwards. We have amended the text accordingly. |
|
|
|
8- Page 4, line 134: unnecessary to refer to Dr. Drew Provan here. Please delete “coordinated by Dr. Drew Provan» |
|
We agree with this comment and the reference to Dr Drew Provan has been omitted. |
|
|
|
9- Page 4, line 154: Corticosteroids provide remissions in 20-30%. I believe 20% is a too low. |
|
Thank you for this comment. I think what was meant by the text was that following discontinuing corticosteroids the remission rate is around 15 to 20%. This means that 80% of patients may require a second-line therapy. In terms of response to corticosteroids, the majority of patients (80%+) will respond to these agents. We hope this clarifies the point. |
|
|
|
10- Page 4, line 164: we recommend limiting the use of cs to 6-8 weeks. That is not primarily because of a negative impact on QoL because we do not know that! I would rather say to limit side effects (short and long term). |
|
Thank you for raising this issue with corticosteroids. There are certainly cases where patients have been treated with corticosteroids for six months or longer with definite impact on quality-of-life including osteoporosis and many other complications. The six week target was chosen in the guidelines because this would minimise the need for any bone protection, and would minimise long-term complications from continuous use corticosteroids. Recognising that the majority of patients will relapse following discontinuation it made better sense when the guidelines were written to minimise exposure to around six weeks to let the patient get through the acute episode and then decide what to do once the corticosteroid effect has worn off. |
|
|
|
11- Page 5: Is there any data showing that romiplostim 10ug is as effective and as quick as IVIG? |
|
Thank you for this point. We are not aware of any specific data showing that the highest dose romiplostim is as effective or quick as IVIG. In the clinical scenarios where this dosage of Romiplostim is used along with IVIG is in the extremely urgent setting where the patient is bleeding extensively and multiple agents are often given in an attempt to raise the platelet count very quickly. |
|
|
|
12- Page 5 line 159. There are no data to support the notion that early use of fosta improves Qol, neither do for TPO. Please revise this statement. |
|
Thank you for this comment. We agree that there are few data for QoL with fosta and TPO-RAs although do have considerable data for TPO overall in terms of better quality-of-life than the older immunosuppressive therapies (I-WISh studies, for example). We have modified the text accordingly |
|
|
|
13- Page 5, line 172: The problem with choosing fosta in patients who fail to respond adequately to corticosteroids and in particular those with high risk of thromboembolism is not always possible because of the limitations imposed by the label (chronic ITP). This has to be clarified. Also why not rituximab to those with high risk of thromboembolism. Please discuss |
|
Thank you for raising these interesting points. We agree about the label being for chronic ITP but in reality we often use these agents outside of the label and certainly earlier than 12 months. Rituximab is not an unreasonable option although our use of rituximab has been very low over the last three years largely driven by the pandemic. However, even before the pandemic, the use of rituximab had declined dramatically largely because of the availability of approved non-immunosuppressive therapies like TPO-TAs. The long-term response with rituximab is not particularly high if you look at the five year data where it looks similar to placebo. But there is definitely still a role for rituximab in the management of ITP. |
|
|
|
14- The authors acknowledge that TPO may cause myelofibrosis. What is their approach/practice to follow up the patients on TPO to discover myelofibrosis. |
|
Thank you for this interesting comment. The term myelofibrosis implies some form of myeloproliferative disease whereas what we see with TPO-RAs is an increase in bone marrow reticulin which is present already in two thirds of patients with ITP before TPO-RA therapy. We have tried to use the term “bone marrow fibrosis” in the review. Our practice is to discontinue the drug and in our experience the increased bone marrow reticulin resolves over time. |
|
|
|
15- Page 6, I cannot see how the 7-8 line describing the Spanish eltrombopag registry and the publications generated from that registry is relevant to the context of the manuscript. The whole passage can be deleted. |
|
Thank you for this comment. The reviewer is quite correct in that the Spanish data and the publications arising from this do not add greatly to the manuscript so we will remove this. |
|
|
|
16- Page 6, line 209: The statement concerning switching to another TPO because of toxicity is not always true. You will not switch to another TPO if the reason was MF. You switch if the AE is related to the drug but you do not do so if the AE is a class effect. This has to be clarified. |
|
We agree with the reviewer here. If a TPO receptor agonist did cause fibrosis of the bone marrow we would not switch to a different TPO receptor agonist. We would use another class of therapy. We have amended the manuscript accordingly |
|
|
|
17- Avatrombopag was recently licensed in Spain…) has unnecessarily been repeated several times. |
|
Thank you for pointing this out the text relating to this throughout the document to prevent repetition. |
|
|
|
18- Page 6, line 230. As mentioned earlier, I would say, there are indications that fosta is effective when used as second line since the evidence was based on post-hoc analysis. |
|
Indeed, data on the efficacy of second-line fostamatinib are obtained after the fact from the studies that led to the approval of fostamatinib for use in ITP. Until a specific study of second-line fostamatinib is developed, questions remain about the efficacy of fostamatinib in this setting. However, the excellent results obtained in this post hoc analysis of data from the phase 3 study of 32 patients who received fostamatinib as second-line therapy demonstrated that a higher proportion of these patients achieved an overall response compared to those who received fostamatinib as last-line therapy (78% vs. 47%). Responses were similarly maintained in patients receiving fostamatinib as second-line and late-line therapy (median 83% and 86% of treatment days, respectively). Thus, these data allow us to recommend the use of fostamatinib in second-line therapy rather than using fostamatinib in later lines of treatment. |
|
|
|
19- Page 6, line 240: please add the rate “rate of XX%” ! |
|
Thank you for spotting this. We have added the number here. |
|
|
|
20- I disagree with the author concerning their opinion on rituximab and in particularly when compared with fosta. Rituximab is more effective than fosta and much better tolerated. I am sure if we had conducted a head-to-head comparison, rituximab would have surpassed fosta in effect, safety and QoL and of course cost-effectiveness. This may also be applies to MMF. The British has extensive experience with MMF. Please moderate your statements regardless of what the Spanish guideline es says. |
|
Thank you for these comments. We agree in terms of MMF which is a useful and gentle second or third line therapy for ITP with good results. It has also been used first line as shown by the Flight trial although adoption as a first-line agent has not followed. Rituximab has a long track record and we have used this off-label for 20 years. We will modify the manuscript as best we can but I think with treatment choices such as these there is always the need to tailor everything to the specific patient. |
|
|
|
21- Unmet needs, first point. I feel there is a mix between diagnostic and prognostic markers. Pls re-phrase to clarify. |
|
Thank you. This has been rephrased in the manuscript. |
|
|
|
22- The statement regarding the use of rituximab during the covid was contraindicated is “outdated” and we know now is incorrect. We all know now that rituximab did not increase the risk of acquiring COVID infection neither did it complicate the course of disease. Even it is impact on suppression of humoral response to vaccine is of unknown clinical value! |
|
Thank you for this comment. Overall throughout the pandemic I think we avoided pretty much every immunosuppressive agent and used TPO-RAs extensively. Our patients who were already on MMF and other immunosuppressants were continued on those rather than risk relapse but starting new immunosuppression was very uncommon and has remained low since the beginning of 2020. The issue with rituximab apart from the immunosuppressive features is the response to vaccination which we do know is suboptimal following rituximab, and during the pandemic there was concern regarding vaccination and booster vaccinations which I think is why we tried to avoid rituximab as far as possible. |
Reviewer 2 Report
ITP, Our Personal View
Tomás JoséGonzález-López1,*, Adrian Newland2 and Drew Provan2.
The authors express their personal views on the diagnosis and treatment of ITP. Guidelines are given for the correct management of ITP in different clinical situations. Given that the authors are among the world's recognized top expert on ITP, their personal opinion in this area makes them valuable.
General comments: The article is unfortunately written too superficially. It lacks many important details, especially about anamnesis, incidence, diagnostics, and underlying pathology.
I miss several obvious references: a. for incidence b. for correlation with age c. for cellular immunity (now a childhood ITP paper is referred to) d. for Th ratio’s e. for TPO and MK involvement (now for both reference 1 is used, but this is not doing the job), etc.
I think it would be better to stick with adult (chronic) ITP, being the obvious autoimmune disorder and not include early childhood ITP. The diagnostic and therapeutic approach for both disorders are completely different. Maybe for you this is obvious, but I think better to state in introduction. Also because of this, reference 6 should not be used
Several paragraphs in the chapter ‘our view on the diagnosis of ITP’ (page 3, line 68-97) are a mix of possible underlying pathology and treatment. Better to structure the paper accordingly. However, in my opinion including the paragraph ‘underlying pathology’ asks for more details and better references.
More specific comments:
Introduction p2 line 51 -53
In our opinion, recommendations from the two standard guidelines are sufficient to establish the diagnosis in the great majority of cases.
Please specify “standard guidelines”. The international and ASH guidelines are the most frequently used guidelines, but are not considered the standard all over the world. Since 2 of the 3 authors participated in the international guideline, it is important to clarify in this article why, for example, a bone marrow examination is waived and McMaster's extensive diagnostic list is not adopted
P2, line 55
Nevertheless, new tools with a potential use for ITP diagnosis are currently being developed. Indeed, we recently submitted our results using next generation flow (NGF) and ITP demonstrating the efficacy of NGF to differentiate ITP and MDS in both bone marrow and peripheral blood samples to the 2022 Congress of the European Hematology Association (EHA) 5.
The results of NGF are not shown and incompletely discussed. We invite the authors to further explore and explain this interesting topic. Now, for the reader, it is no different than an announcement
ITP is primarily characterized by platelet and megakaryocytic destruction caused by antiplatelet autoantibodies present in the serum and bone marrow of the patient. These autoantibodies are observed free and also bound to platelets2. These anti-platelet antibodies, even with high sensitivity and specificity, can only be detected in sera from around 60% of patients. In the remaining 40% of cases, antibodies are not present but their presence is not required for the diagnosis of ITP. In addition, the presence or absence of antiplatelet antibodies or their subtypes are not correlated with treatment response. Thus, we do not support routine testing for anti-platelet antibodies since our treatment results are similar independently of them.
Your view on diagnostics is confusing. In your opinion recommendations from the two ‘standard’ guidelines are in the majority of cases sufficient. However, you recommend NGF to differentiate ITP from MDS and in your ‘unmet needs’ point 1 starts with diagnosis of ITP remains a very important challenge. I agree and therefore I support the opinion of Kelton and Vrebensky to include patient platelets-associated glycoprotein specific autoantibody detection (at least for patients for whom it is not a straightforward diagnosis) as a rule-in test. Not only tailoring therapy (line 118), but also tailoring diagnostics should be recommended. For completely rejecting autoantibody detection as potentially useful, your paragraph about autoantibody detection (page 2) is insufficient. A 2017 reference is used (ref 2) which is not up to date. Better to refer to Kelton (ref 4 in your paper) and Porcelijn (Transfus Med Rev. 2020) You mention … detected in sera from about 60% of … This should be on patient platelets and varying between 50 and 80%. In the McMaster registry, 12% is initially not correctly ITP diagnosed. Meaning, a reliable rule-in test (specificity of the glycoprotein specific assay is >90%) is welcome. Furthermore, not correlated with treatment response is possibly true, but not sufficiently investigated and at least relevant refs should be included.
I also miss (especially from these authors) some comments on the place of autologous platelet sequestration investigation.
P3, line 80:
In chronic ITP there is also an alteration in the Th1/Th2 ratio in favor of a Th1 response (proinflammatory) which leads to an increase in Th1 precursors and to the observation of a specific cytokine profile (very high IL-2 and IFN
To the best of our knowledge, no new clinical trial drugs aim to modify these cellular and cytokine alterations although recent discoveries support that suggestion that some agents, such as thrombopoietin receptor agonists (TPO-RAs), may attain a platelet response due, in part, to a modification of this cellular and humoral imbalance. In our opinion, it is essential that we fined find new drugs that target these cellular alterations especially when we are dealing with refractory ITP. Currently for these patients combination therapies are still the only therapeutic solution available.
We fully support the authors' opinion that cellular abnormalities should also be an important target in the treatment of ITP. However, we miss the discussion of longstanding immunosupressants, of which MMF is an example for the first-line treatment of ITP: Mycophenolate Mofetil for First-Line Treatment of Immune Thrombocytopenia, Charlotte A. Bradbury, M.D., Ph.D. N Engl J Med 2021; 385:885-895
P3, line 94
e.g. fostamatinib, that may prevent the destruction or phagocytosis at the splenic level of platelets opsonized with autoantibodies8. We are encouraged by the therapeutic results that this type of drug can provide us in the treatment of our patients with ITP. Our early clinical results support its use in the refractory patient with optimal data in the second line scenario.
There are still insufficient data for the use of fostamatinib in the second line. Can the authors indicate their personal opinion for the use of fostamatinib in relation to other second-line agents such as the TPO-RA and Rituximab. Do the authors advise e.g. after rituximab, to use multiple TPO-RAs first, or try combination therapy with a TPO-RA and an immunosupressant?
As previously mentioned, we consider TPO-RAs and fostamatinib the most effective and safe drugs approved for second-line ITP nowadays3,9.
To our knowledge, fostamatinib is registered as a third-line treatment in ITP in most EU countries. The drug has an overall efficacy of 40% and should be used continuously. Rituximab has a much milder side effect profile and reasonable efficacy. TPO-RA and Rituximab are the second-line agents of first choice and not fostamatinib. We ask the authors to better justify their therapeutic choice.
Page 4 line 156-161
Your opinion on IVIg in ITP is not clear. First you mention IVIg for those patients who are bleeding or at high risk of bleeding. Later you prefer romiplostim for emergency situations.
P6, Line 235-240: 80% early response rate for splenectomy is mentioned, but this should be nuanced with figures for the long term. The percentage XX% (line 240) should be filled in.
P7, line 246
Outside of this pandemic, current Spanish guidelines recommend these drugs as third-line treatment options after the use of corticosteroids, intravenous immunoglobulins, thrombopoietin analogues (TPO-RAs; romiplostim, eltrombopag, avatrombopag) and Syk inhibitors (fostamatinib)27. We also consider rituximab and other immunosuppressants such as cyclophosphamide, danazol, cyclosporine, mycophenolate mofetil, azathioprine, vinca alkaloids and/or dapsone rather outrated and only to be used in third or further lines for highly refractory patients9.
This proposed therapeutic policy is completely opposite to what the international developments suggest. On the contrary, the use of rituximab and other immunosuppressants early in treatment is increasingly being promoted. See also our previous reference on MMF in first-line treatment
P7, line 272
Unmet need 3: I do not agree with the remark that we lack assays to detect variation in the aspects mentioned. It is possible to investigate all of these aspects, however, the assays are complicated and not for routine use.
Overall conclusion:
Although the opinion of experts in the field of diagnosis and treatment of ITP is very valuable, this article lacks a clear structure and good rationale for the diagnostic and therapeutic choices made. Some advice also goes against the advice given in the international guideline, which Provan and Newland wrote themselves. This makes this article appear very confusing to the average ITP practitioner. We therefore think that this article should not be considered for publication in this form.
Author Response
|
ITP, Our Personal View |
|
REVIEWER #2 |
|
Tomás JoséGonzález-López1,*, Adrian Newland2 and Drew Provan2. |
|
The authors express their personal views on the diagnosis and treatment of ITP. Guidelines are given for the correct management of ITP in different clinical situations. Given that the authors are among the world's recognized top expert on ITP, their personal opinion in this area makes them valuable. |
|
|
|
General comments: The article is unfortunately written too superficially. It lacks many important details, especially about anamnesis, incidence, diagnostics, and underlying pathology. I miss several obvious references: a. for incidence b. for correlation with age c. for cellular immunity (now a childhood ITP paper is referred to) d. for Th ratio’s e. for TPO and MK involvement (now for both reference 1 is used, but this is not doing the job), etc. I think it would be better to stick with adult (chronic) ITP, being the obvious autoimmune disorder and not include early childhood ITP. The diagnostic and therapeutic approach for both disorders are completely different. Maybe for you this is obvious, but I think better to state in introduction. Also because of this, reference 6 should not be used Several paragraphs in the chapter ‘our view on the diagnosis of ITP’ (page 3, line 68-97) are a mix of possible underlying pathology and treatment. Better to structure the paper accordingly. However, in my opinion including the paragraph ‘underlying pathology’ asks for more details and better references. |
|
We thank the reviewer for these helpful comments. We agree that some areas were too superficial, and others were probably too long and muddled (e.g. pathophysiology). We have tightened the whole manuscript by restricting our comments to adult primary ITP, minimal pathophysiology with additional references for those interested in reading about this in detail. The important work of Frederiksen to which the reviewer is alluding has been added since the age distribution in ITP is important. We agree with the reviewer also that mixing in the pathophysiology with therapeutic approach has made the review muddled but we have separated these out now. |
|
|
|
More specific comments: Introduction p2 line 51 -53 |
|
In our opinion, recommendations from the two standard guidelines are sufficient to establish the diagnosis in the great majority of cases. |
|
Please specify “standard guidelines”. The international and ASH guidelines are the most frequently used guidelines, but are not considered the standard all over the world. Since 2 of the 3 authors participated in the international guideline, it is important to clarify in this article why, for example, a bone marrow examination is waived and McMaster's extensive diagnostic list is not adopted |
|
Thank you. We agree with the comments. ICR and ASH are large guidelines but there are several country-specific guidelines also. We have addressed the bone marrow issue and added a relevant citation for this. |
|
|
|
P2, line 55 |
|
Nevertheless, new tools with a potential use for ITP diagnosis are currently being developed. Indeed, we recently submitted our results using next generation flow (NGF) and ITP demonstrating the efficacy of NGF to differentiate ITP and MDS in both bone marrow and peripheral blood samples to the 2022 Congress of the European Hematology Association (EHA) 5. The results of NGF are not shown and incompletely discussed. We invite the authors to further explore and explain this interesting topic. Now, for the reader, it is no different than an announcement |
|
Thank you. We agree that this paragraph was vague and unhelpful. This has been rewritten and citations added. We have also made it clear that NGF is not a mainstream investigation for the diagnosis of ITP at present. |
|
|
|
ITP is primarily characterized by platelet and megakaryocytic destruction caused by antiplatelet autoantibodies present in the serum and bone marrow of the patient. These autoantibodies are observed free and also bound to platelets2. These anti-platelet antibodies, even with high sensitivity and specificity, can only be detected in sera from around 60% of patients. In the remaining 40% of cases, antibodies are not present but their presence is not required for the diagnosis of ITP. In addition, the presence or absence of antiplatelet antibodies or their subtypes are not correlated with treatment response. Thus, we do not support routine testing for anti-platelet antibodies since our treatment results are similar independently of them. |
|
Your view on diagnostics is confusing. In your opinion recommendations from the two ‘standard’ guidelines are in the majority of cases sufficient. However, you recommend NGF to differentiate ITP from MDS and in your ‘unmet needs’ point 1 starts with diagnosis of ITP remains a very important challenge. I agree and therefore I support the opinion of Kelton and Vrebensky to include patient platelets-associated glycoprotein specific autoantibody detection (at least for patients for whom it is not a straightforward diagnosis) as a rule-in test. Not only tailoring therapy (line 118), but also tailoring diagnostics should be recommended. For completely rejecting autoantibody detection as potentially useful, your paragraph about autoantibody detection (page 2) is insufficient. A 2017 reference is used (ref 2) which is not up to date. Better to refer to Kelton (ref 4 in your paper) and Porcelijn (Transfus Med Rev. 2020) You mention … detected in sera from about 60% of … This should be on patient platelets and varying between 50 and 80%. In the McMaster registry, 12% is initially not correctly ITP diagnosed. Meaning, a reliable rule-in test (specificity of the glycoprotein specific assay is >90%) is welcome. Furthermore, not correlated with treatment response is possibly true, but not sufficiently investigated and at least relevant refs should be included. |
|
Thank you. These are very useful points. The issue of anti-platelet antibody testing comes up often with clinicians, and Porcelijn has published very useful data. We ourselves use MAIPA occasionally and we describe this in the text. We were trying to convey our practices and in general, anti-platelet antibody testing is not available everywhere and standardisation/quality control varies from department to department. Their role may become clearer and feature in future guidelines, but we have acknowledged their utility. However, we would not, at present, recommend they be done routinely in ITP management. |
|
|
|
I also miss (especially from these authors) some comments on the place of autologous platelet sequestration investigation. |
|
Thank you. We are not sure if the reviewer is referring to Indium labelled scanning which is caried out in a few centres worldwide. In London we do perform this test but only when we are considering splenectomy. We do not use it in a diagnostic manner. |
|
|
|
P3, line 80: |
|
In chronic ITP there is also an alteration in the Th1/Th2 ratio in favor of a Th1 response (proinflammatory) which leads to an increase in Th1 precursors and to the observation of a specific cytokine profile (very high IL-2 and IFN |
|
We fully support the authors' opinion that cellular abnormalities should also be an important target in the treatment of ITP. However, we miss the discussion of longstanding immunosupressants, of which MMF is an example for the first-line treatment of ITP: Mycophenolate Mofetil for First-Line Treatment of Immune Thrombocytopenia, Charlotte A. Bradbury, M.D., Ph.D. N Engl J Med 2021; 385:885-895 |
|
We thank the reviewer for highlighting this area. Like many hematologists we had been reducing the use of immunosuppression (IS) over the last few years. The pandemic severely limited IS use and the availability of three TPO-RAs allowed for a different model of ITP management, where finance allowed. We accept that many countries have insufficient access to, or funds for, early use TPO-RAs. IS has been the mainstay for ITP management for decades and there are some useful agents. We agree that MMF appears useful and the Flight trial, with MMF first line, showed some interesting data. Nonetheless MMF is not approved for ITP, the quality of life in the MMF arm of the study was less good and because of the pandemic (infection concerns) MMF is seldom used first line. We do use MMF but not in the first line currently. We have amended the text to reflect our practice regarding IS use. |
|
|
|
P3, line 94 |
|
e.g. fostamatinib, that may prevent the destruction or phagocytosis at the splenic level of platelets opsonized with autoantibodies8. We are encouraged by the therapeutic results that this type of drug can provide us in the treatment of our patients with ITP. Our early clinical results support its use in the refractory patient with optimal data in the second line scenario. |
|
There are still insufficient data for the use of fostamatinib in the second line. Can the authors indicate their personal opinion for the use of fostamatinib in relation to other second-line agents such as the TPO-RA and Rituximab. Do the authors advise e.g. after rituximab, to use multiple TPO-RAs first, or try combination therapy with a TPO-RA and an immunosupressant? |
|
Thank you. We agree with the reviewer here. Our practice is to use all three TPO-RAs if necessary and if we have complete failure after trying all three, we would consider rituximab or fostamatinib. As the reviewer mentions in the next point, the efficacy of fostamatinib overall is quite low and does require continuous use. Fostamatinib also has side effects for some patients which we have added to the text. Rituximab has been used for almost 20 years in ITP and has a good track record although the long term rates of remission are probably similar to placebo. We have addressed this in the text providing guidance in terms of the order in which we use the various agents which we hope will be helpful. |
|
|
|
As previously mentioned, we consider TPO-RAs and fostamatinib the most effective and safe drugs approved for second-line ITP nowadays3,9. |
|
To our knowledge, fostamatinib is registered as a third-line treatment in ITP in most EU countries. The drug has an overall efficacy of 40% and should be used continuously. Rituximab has a much milder side effect profile and reasonable efficacy. TPO-RA and Rituximab are the second-line agents of first choice and not fostamatinib. We ask the authors to better justify their therapeutic choice. |
|
Thank you for this comment. We agree that fostamatinib would be considered a third line therapy and our practice would be to try all three TPO-RAs before moving to fostamatinib therapy; the efficacy is not particularly high and it does have to be given continuously. To date our practice has been largely to reserve fostamatinib for patients who fail multiple therapies but this will depend on the individual patient. We have provided of the sequencing of therapies in the manuscript which we hope will be helpful. |
|
|
|
Page 4 line 156-161 |
|
Your opinion on IVIg in ITP is not clear. First you mention IVIg for those patients who are bleeding or at high risk of bleeding. Later you prefer romiplostim for emergency situations. |
|
In terms of IVIg we reserve this for patients who are bleeding or high risk of bleeding. The actual dosing schedule varies depending on the country. Some years ago IVIg was used as a maintenance therapy but we now reserve it for emergencies/rescue or sometimes prior to surgical or dental procedures. We also use IVIg in pregnancy. We have outlined this more fully in the manuscript which we hope will be satisfactory. |
|
|
|
P6, Line 235-240: 80% early response rate for splenectomy is mentioned, but this should be nuanced with figures for the long term. The percentage XX% (line 240) should be filled in. |
|
Thank you for this comment regarding splenectomy. The reviewer is quite correct that the initial early response is quite high but over time some patients will relapse and the five years to response will be around 60%. We have modified the paper accordingly.
We have also amended the “XX%”. Thank you for alerting us to this error.
|
|
|
|
P7, line 246 |
|
Outside of this pandemic, current Spanish guidelines recommend these drugs as third-line treatment options after the use of corticosteroids, intravenous immunoglobulins, thrombopoietin analogues (TPO-RAs; romiplostim, eltrombopag, avatrombopag) and Syk inhibitors (fostamatinib)27. We also consider rituximab and other immunosuppressants such as cyclophosphamide, danazol, cyclosporine, mycophenolate mofetil, azathioprine, vinca alkaloids and/or dapsone rather outrated and only to be used in third or further lines for highly refractory patients9. |
|
This proposed therapeutic policy is completely opposite to what the international developments suggest. On the contrary, the use of rituximab and other immunosuppressants early in treatment is increasingly being promoted. See also our previous reference on MMF in first-line treatment |
|
Thank you for this comment. Our view is that worldwide there has been a gradual reduction on immunosuppressive agent use for ITP. This has been even more dramatic since the pandemic and the introduction of less immunosuppressive therapies. Our use of rituximab has dropped significantly also and we have used little rituximab since the beginning of 2020.
We have rephrased the description of rituximab and immunosuppression to suggest that they may be used on a patient-by-patient basis depending on the circumstances, prior therapies and patient preference.
|
|
|
|
P7, line 272 |
|
Unmet need 3: I do not agree with the remark that we lack assays to detect variation in the aspects mentioned. It is possible to investigate all of these aspects, however, the assays are complicated and not for routine use. |
|
Thank you for this comment. In terms of Unmet need 3, we would like to retain this as an unmet need because we do not truly know which patients will respond to each of the specific treatments for ITP which makes it difficult to select specific therapies for individuals. And although we have a better understanding of the underlying pathophysiology of ITP in general, within a specific patient we do not know whether this is predominantly B cell, or T cell, NK cell, monocyte, DC, or complement-mediated. There may be some research assays available which could help here but in terms of clinical utility we feel it will be some years until this level of sophistication will be available. For this reason, we feel that Unmet need number 3 should be left in.
|
|
|
|
Overall conclusion: |
|
Although the opinion of experts in the field of diagnosis and treatment of ITP is very valuable, this article lacks a clear structure and good rationale for the diagnostic and therapeutic choices made. Some advice also goes against the advice given in the international guideline, which Provan and Newland wrote themselves. This makes this article appear very confusing to the average ITP practitioner. We therefore think that this article should not be considered for publication in this form. |
|
|

Reviewer 3 Report
Please refer to the file attached. Thanks

Author Response
|
ITP, Our Personal View |
|
REVIEWER #3 |
|
|
|
This manuscript is of a very high interest for the readers. However a number of major issues must be resolved prior to its publication in the MDPI Medicina journal. Additionally, extensive editing of English language, sentence structure, and style are required to improve the readability and clarity of the research. The major issues are outlined below alongside many minor errors that distract from reading and do not allow to fully appreciate the soundness of this viewpoint piece. |
|
|
|
Title needs to be more descriptive. It will benefit from including some of key words – diagnosis, management, treatment, guidelines – to reflect what aspect of ITP is of interest. |
|
Thank you. We agree with the reviewer. The title has been changed to make it clearer and more descriptive. |
|
|
|
Abstract: rather than summarizing the manuscript or highlighting why this research is important, the abstract is comprised of copy-pasted sentences from the introduction section. |
|
Thank you. We have revised the abstract making it less cut-and-paste which hopefully will make it much more useful. |
|
|
|
The word is missing line 11: “higher in … than adults”. |
|
We thank the reviewer for spotting this. We have reworded the summary so that no words are missing in line 11. |
|
|
|
Check punctuation and sentence structure to improve clarity here and in the manuscript body. |
|
Thank you. We agree completely with the reviewer’s comments here. The entire manuscript has been edited and the English tidied substantially which will hopefully make it more readable. |
|
|
|
Key words: too vague. Remove: molecular biology and cytometry. |
|
Thank you. We have revised the keywords removing vague terms. |
|
|
|
Introduction: Include brief introductory information on pathophysiology and known molecular mechanisms involved of the disease. It is vital in order to understand other pieces of information randomly introduced in the next section “Our view on the diagnosis”: cellular immunity, production of autoantibodies, what’s TH1/TH2, why we should care about spleen tyrosine kinase, what’s TRO/Ras etc. |
|
We agree with the reviewer. The pathophysiology of ITP is a huge topic and probably a detailed discussion is outside the scope of this short review. Many good reviews on the underlying mechanisms have been published. We have reduced this section providing only an outline of the pathophysiology which we hope will be useful. Readers interested in the detailed pathophysiology can consult the reviews cited in the text for a more authoritative discussion on the known mechanisms. |
|
|
|
Authors offer classification primary vs secondary ITP, but do not brief the reader on what is refractory ITP (line 297), multirefractory ITP, and second-line ITP (line 311) mentioned in conclusions. |
|
Thank you. The terminology in ITP is confused generally, with disagreement among experts about what the words mean, (note there will be a new consensus document on terminology which will redefine “refractory” and other confusing terms). We have avoided discussing refractory/multi-refractory and instead focus on adult primary ITP only, which we define early on. We have avoided the paediatric issues and terminology controversy. |
|
|
|
Our view on the diagnosis: |
|
Please explain what’s a “diagnosis of exclusion” – line 46. |
|
Thank you. We have now defined this in the text. |
|
|
|
Describe what efforts are currently made for early diagnostic of ITP. The conclusion is made about the usability of flow cytometry for the diagnosis, but no information is presented anywhere in the text except referencing the sentence referencing authors abstract (lines 56-58). New generation flow cytometry – what is this? Conventional multi-colored flow cytometry or imaging flow cytometry. Please stick to the terminology familiar to the general audience of the readers. |
|
Thank you. We agree this should have been discussed in greater detail. Originally we provided a vague statement about NGF but this has been described better and supporting references have been added to help the reader. |
|
|
|
Line 60 and further, when authors say ITP, do they refer to primary, secondary, or both? |
|
Thank you. We are dealing in this review solely with Primary ITP in adults. Secondary ITP has not been discussed, to avoid confusion. |
|
|
|
Random pieces on the ITP pathology and molecular biology are introduced here without clear connection of one to another: cellular immunity, cytokines, tyrosine kinase pathways, etc. Please offer an introduction explaining why this info is important for the diagnosis of ITP section. Consider introducing subheadings. |
|
We agree with the reviewer here. As mentioned above, the pathophysiology is a large and complex topic and, as written, was disjointed. We have slimmed this down giving only salient points.
|
|
Strangely, no information is provided regarding the mechanisms of platelet elimination during ITP, as the primary cause of lack of platelets in bloodstream. |
|
Thank you. We did mention the under-production issue at the beginning of the review as a contributing factor to the thrombocytopenia. We have amended some of the text here to make it clearer. |
|
|
|
Please carefully define all abbreviations in the text, including TH1/TH2, IFN. |
|
Thank you for pointing this out. All abbreviations have now been given in full when mentioned the first time. |
|
|
|
Avoid extra lengthy references (line 55-57, 115, 128) to your research and stick to widely accepted format, for example “as discussed in our previous manuscript”. Do not include the information that you have published the manuscript “in high-impact peer-review journal” (line 128). It is rather obvious for the reader. |
|
We agree with the statement, and we have removed references to previous works or redundant statements. |
|
|
|
Conclusions: only partially reflect the information introduced in the manuscript. Some conclusions appear out of nowhere, and the topics are not discussed in this particular report. The topics include (1) new generation flow cytometry – what is this? Conventional multi-colored flow cytometry or imaging flow cytometry. Please stick to the terminology familiar to general audience. (2) No discussion in the text is available on refractory ITP and 2nd line ITP. |
|
We agree that the conclusions did not reflect the text well. Hopefully with the extensive revision we have addressed this. |
|
Considering all the above, the major revision of the manuscript is mandatory. Thanks |

Round 2
Reviewer 1 Report
This revised version has improved substantially. The authors have responded adequately to my comments and suggestions.
Author Response
This revised version has improved substantially. The authors have responded adequately to my comments and suggestions.
We are pleased to have responded accurately to reviewer 1’s comments and suggestions.

Reviewer 2 Report
Unfortunately, with the exeption of more references and some structural adjustments, the authors have done too little with te comments made earlier.
The article remains superficial and based in my opinion on insufficiently supported opinions.
Please look again at the previous comments, of which some are:
The section on autoantibody detection still shows little knowledge. I am not concerned with the opinion whether or not autoantibody detection has a place in the routine diagnosis of ITP, but with the way in which this is presented. The authors do not seem to understand the difference between autoantibody detection on the patient's platelets or in the serum. Antibody detection in the serum only has a sensitivity of around 40%, but on the patient's platelets 50-80%. The high specificity of the glycoprotein specific antibody detection assay, which can be of diagnostic use, is not mentioned.
The comment in "the take home message", that Arnold's recommendations may be difficult for most centers: perhaps the authors might consider referral of ITP patients (at least for a good initial diagnostic work-up) to better equipped centers?
The different therapeutic options should be supported by response rates.
The comment about rituximab treatment during the covid-19 period is too suggestive and should be substantiated more. As far as I understand, the GEPTI text (translated in Google) states that, given the lack of sufficient knowledge, one should be careful with immunosuppressants because there is a risk of more serious Covid-19 infections. This recommendation was written in 2020. More is now known, such as the article by Levavi (An of Hematology, 2021), which shows that there is no evidence of adverse effects.
Fostamatinib is one of the options for (refractory) ITP patients, but the sustained response rates are worse than reported here. A better paragraph has been written about this in the updated international consensus (Provan et al), repeatedly cited by the authors.
I still
Author Response
REVIEWER #2
Unfortunately, with the exeption of more references and some structural adjustments, the authors have done too little with te comments made earlier.
The article remains superficial and based in my opinion on insufficiently supported opinions.
Please look again at the previous comments, of which some are:
The section on autoantibody detection still shows little knowledge. I am not concerned with the opinion whether or not autoantibody detection has a place in the routine diagnosis of ITP, but with the way in which this is presented. The authors do not seem to understand the difference between autoantibody detection on the patient's platelets or in the serum. Antibody detection in the serum only has a sensitivity of around 40%, but on the patient's platelets 50-80%. The high specificity of the glycoprotein specific antibody detection assay, which can be of diagnostic use, is not mentioned.
Thank you for this comment. The reviewer is quite correct and we have modified the text accordingly mentioning the two statements that reviewer 2 has commented.
The comment in "the take home message", that Arnold's recommendations may be difficult for most centers: perhaps the authors might consider referral of ITP patients (at least for a good initial diagnostic work-up) to better equipped centers?
Thank you for your kind suggestion. We have now deleted the first statement and, on the other hand, we have included the suggestion to consider referring ITP patients (at least for a good initial diagnostic evaluation) to better equipped centres.
The different therapeutic options should be supported by response rates.
Very interesting point. Really appreciate your comment. Now we have included the response rates from different therapies in the text.
The comment about rituximab treatment during the covid-19 period is too suggestive and should be substantiated more. As far as I understand, the GEPTI text (translated in Google) states that, given the lack of sufficient knowledge, one should be careful with immunosuppressants because there is a risk of more serious Covid-19 infections. This recommendation was written in 2020. More is now known, such as the article by Levavi (An of Hematology, 2021), which shows that there is no evidence of adverse effects.
Thank you again for raising this interesting point. We now mention in our manuscript the paper from Levavi reporting the no evidence of adverse events when using immunosuppressants even in the Covid-19 pandemic scenario.
Fostamatinib is one of the options for (refractory) ITP patients, but the sustained response rates are worse than reported here. A better paragraph has been written about this in the updated international consensus (Provan et al), repeatedly cited by the authors.
We agree with the reviewer here. Sustained response rates of fostamatinib are worse than reported here so we have modified this data in the text. Furthermore, we also agree that a better paragraph was written about this in 2019 in the updated international consensus (Provan et al) and, thus, we have modified our manuscript accordingly.

Round 3
Reviewer 2 Report
No further comments